# The Significance of NK1 Receptor Ligands and Their Application in Targeted Radionuclide Tumour Therapy

**DOI:** 10.3390/pharmaceutics11090443

**Published:** 2019-09-01

**Authors:** Agnieszka Majkowska-Pilip, Paweł Krzysztof Halik, Ewa Gniazdowska

**Affiliations:** Centre of Radiochemistry and Nuclear Chemistry, Institute of Nuclear Chemistry and Technology, Dorodna 16, 03-195 Warsaw, Poland

**Keywords:** neurokinin 1 receptor, Substance P, SP analogues, NK1R antagonists, targeted therapy, radioligands, tumour therapy, PET imaging

## Abstract

To date, our understanding of the Substance P (SP) and neurokinin 1 receptor (NK1R) system shows intricate relations between human physiology and disease occurrence or progression. Within the oncological field, overexpression of NK1R and this SP/NK1R system have been implicated in cancer cell progression and poor overall prognosis. This review focuses on providing an update on the current state of knowledge around the wide spectrum of NK1R ligands and applications of radioligands as radiopharmaceuticals. In this review, data concerning both the chemical and biological aspects of peptide and nonpeptide ligands as agonists or antagonists in classical and nuclear medicine, are presented and discussed. However, the research presented here is primarily focused on NK1R nonpeptide antagonistic ligands and the potential application of SP/NK1R system in targeted radionuclide tumour therapy.

## 1. Introduction

Neurokinin 1 receptor (NK1R), also known as tachykinin receptor 1 (TACR1), belongs to the tachykinin receptor subfamily of G protein-coupled receptors (GPCRs), also called seven-transmembrane domain receptors (Figure 1) [1,2,3]. The human NK1 receptor structure [4] is available in Protein Data Bank (6E59). Tachykinins, widely distributed within the central (CNS) and peripheral (PNS) nervous system, are small bioactive neuropeptides which share a conserved C-terminal pentapeptide sequence, Phe-X-Gly-Leu-Met-NH_2_. Examples of these neurotransmitters belonging to the tachykinin group include Substance P (SP), the first neuropeptide discovered in mammals [5], neurokinin A (NKA) and neurokinin B (NKB). These compounds listed above are the preferential ligands for NK1, NK2 and NK3 receptors, respectively, although they can bind additional NK receptors with varying affinity [6].

Within the neurokinin receptor family, there are three pharmacologically distinct receptor subtypes: NK1R (TACR1, SPR), NK2R (TACR2) and NK3R (TACR3) [7,8,9]. NK1R is widely expressed in both the CNS and PNS, whereas NK2R is preferentially expressed in PNS [9]. NK1R contains 407 amino acids, and NK2R and NK3R (the longest one) consist of 398 and 465 amino acids, respectively [3]. NK1R exists in two isoforms, as a full-length peptide (NK1R-Fl, Figure 1) and in the truncated isoform (NK1R-Tr), containing 311 amino acids (96 amino acids less at the C-terminus) [3]. NK1R displays two nonstoichiometric binding sites, the more abundant NK-1M (“majority”—representing 80–85% of the total receptor population) and NK-1m (“minority”—so-called “septide sites” or “septide-sensitive”) defined according to the different binding potencies of SP and its analogues [10,11,12,13,14]. NK1R contains an extracellular N-terminus, three extracellular loops (E1, E2 and E3), seven transmembrane domains, three intracellular loops (C1, C2 and C3, as well as a possible C4 loop) and an intracellular C-terminus.

The wide overexpression of NK1R in various human organs has led to successful development of highly selective agonists and antagonists of this receptor for the treatment of various diseases [8]. Some NK1R ligands, for example SP, its analogues and derivatives, have been investigated in preclinical and clinical studies. Moreover, based on high density of transmembrane NK1Rs on human cancer cells, new therapeutic approaches involve the use of radiolabelled NK1R ligands in targeted radionuclide tumour therapy [2,15,16,17].

The aim of this review is to discuss data from recent literature concerning the chemical and biological aspects of natural and synthetic NK1R ligands in classical and nuclear medicine, with a specific focus on targeted radionuclide therapy.

## 2. NK1R Ligands in Classical Medicine

### 2.1. Significance of NK1R Agonists

The endogenous peptide ligands of NK1R are tachykinins, a large family of neuropeptides produced by neuronal and glial cells [5,18]. These compounds play an important role in nociception, synaptic transmission (as excitatory neurotransmitters) and neuroimmunomodulation. They have a variety of effects on physiological and pathological conditions, as well as intrinsic neuroprotective and neurodegenerative properties [19]. The tachykinin family is characterized by a common C-terminal 5-amino-acid sequence Phe-X-Gly-Leu-Met-NH_2_, where X is either an aromatic (Phe or Tyr) or a branched chain aliphatic (Val or Ile) amino acid residue [19,20,21].

The most widespread and tested agonistic peptide ligand of NK1R is human endogenous undecaneuropeptide Substance P (Arg-Pro-Lys-Pro-Gln-Gln-Phe-Phe-Gly-Leu-Met-NH_2_, SP(1–11), SP, [Arg^1^]SP, Figure 2). At physiological pH, SP characteristically holds a positive charge on N-terminal amino acid residues. The C-terminus contains hydrophobic residues providing the SP peptide with an amphiphilic character [22]. The interaction between SP and NK1R results in internalisation of the membrane-bound complex, via a clathrin-dependent mechanism, to the acidified endosomes, where the complex disassociates [8,22,23,24].

The SP fragment responsible for NK1R affinity is a five amino acid sequence, Phe^7^–Phe^8^–Gly^9^–Leu^10^–Met^11^, located at the C-terminus of the peptide [21,25,26]. Tachykinins are degraded by neutral endopeptidases (NEPs) and the angiotensin-converting enzyme (ACE) [27]. The half-life of SP ranges from seconds to 1.5 h in blood and tissues, while in extracted plasma SP is stable for hours [22]. In the biodegradation process the SP metabolites, SP(1–4), SP(1–5), SP(1–6), SP(1–7), SP(8–9) and SP(10–11), can form as a results of enzymatic peptide cleavage at different sites [19,21,27,28,29,30,31,32,33,34,35,36,37]. Typical cleavage points are Lys–Arg, Arg–Arg and Arg–Lys doublets [19]. SP and NK1R are widely distributed in the CNS and PNS. SP mediates neural–immune/hematopoietic cross-talk [20] in the brain, and has specifically been isolated from the brain regions which regulate emotion (hypothalamus, amygdala and the periaqueductal gray) [19,28]. Additionally, SP plays an important role in the aetiology of many diseases [22]. It is thought that SP has a role in the regulation of depression and anxiety, emesis, pain, asthma, psoriasis, inflammatory bowel disease and in diseases of the CNS, including migraines and schizophrenia. The inhibition of NK1R stimulation via NK1R-specific secondary messenger pathways may be a useful in treatment in a variety of diseases. Concise information concerning the biological activity and the potential application of SP, its analogues and derivatives in medicine are presented in Table 1.

Multiple groups have recently published research concerning SP and its analogues or derivatives as agonistic or antagonistic peptide NK1R ligands [7,8,19,21,38,39,40,41,42,43,44,45,46,47,48,49,50,51,52,53,54,55,56,57,58,59,60,61,62,63,64,65]. New unnatural peptides are designed in order to improve their biological properties (e.g., half-life, receptor affinity and lipophilicity). Moreover, appropriate modifications can give peptides specific properties (e.g., photoactivation) or allow labelling with iodine radioisotopes or with other diagnostic/therapeutic radionuclides. NK1R has also been identified on various cancer cells, including astrocytoma, neuroblastoma, melanoma, and pancreatic cancer cells. NK1Rs have been implicated in tumour tissue growth and SP contributes to cancer volume increase, angiogenesis, proliferation and metastasis [5,49,50]. The authors of the patent “Radiolabeled conjugates based on substance P and the uses thereof” examined about ten different SP radiolabelled analogues (Table 2) in order to select optimal analogues in terms of stability and receptor affinity [53]. Endogenous SP is a potent vasodilator [54] and this action limits the application of SP-derived medical preparations.

The autoradiography studies carried out on tumour tissues expressing NK1Rs revealed that ^111^In-DOTA-[Thi^8^,Met(O_2_)^11^]SP radiobioconjugate exhibited the highest affinity to examined receptors, therefore the SP analogue [Thi^8^,Met(O_2_)^11^]SP was used in further development for the synthesis of radiopharmaceuticals applied in medicinal experiments for recurrent, critically located glioblastoma multiforme (GBM) [55].

The SP derivatives agonistic ligand [Sar^9^,Met(O_2_)^11^]SP(1–11) and antagonistic ligand [Tyr^6^, d-Phe^7^, d-His^9^]SP(6–11) (Sendide), along with others, were recently used to examine the role of NK1R in regulation and release of vasopressin peptide (AVP) from neurohypophysis [56,57]. Notably, it has been reported that replacement of l-amino acid by d-amino acid in the peptide results in antagonizing effects [21,58]. Numerous SP analogues (Table 1) with agonistic or antagonistic properties towards SP have been designed based on results of structure activity relationship (SAR) studies.

There are also non-mammalian NK1R ligands belonging to the tachykinin family. These compounds have distinctly different structure from SP, however in COOH-terminal region they have similar pentapeptide sequence Phe-(Tyr/Ile)-Gly-Leu-Met-NH_2_ (often referred as the “message domain”) [8,19,62,63,64,65] (Table 1). The non-mammalian tachykinins display the same spectrum of activity as the mammalian tachykinins and bind to the same receptors.

Taking into account the above information, it can be concluded that the measurement of the level of NK1Rs agonists (SP and its analogues), may be used as a diagnostic probe to inform about disease state. Nevertheless, literature shows the results of using NK1R agonists in application as drug active substances [19] (Table 1). Despite the existence of several examples of the use of NK1R agonists in therapy, it is anticipated that antagonists will be much more effective therapeutics.

### 2.2. Application of NK1R Antagonists

Despite the wide expression of the NK1R and the implication of SP in physiological regulation, NK1R antagonists remain infrequent in clinical use. Among NK1R antagonists aprepitant (MK-869, L-754,030) has been used to prevent of chemotherapy-induced nausea and vomiting (CINV) and obtained market registration (Emend^®^, Merck) by United States Food and Drug Administration (US FDA) and European Medicines Agency (EMA) in 2003 [3,40,66]. Application in patients during moderate and acute emetogenic chemotherapy (e.g., cisplatin, doxorubicin or cyclophosphamide) was broadened by postoperative nausea and vomiting (PONV) indication two years after registration. In 2008, fosaprepitant (Ivemend^®^, Merck), an intravenous prodrug of aprepitant, obtained FDA and EMA approvals. Aprepitant regimen [67,68,69] shows significantly higher response rates in CINV patients than the standard therapy (5-HT_3_R antagonist ondansetron combined with corticosteroid dexamethasone). The antiemetic effect of aprepitant was reported to be due to the inhibition of high density NK1R brainstem regions (the area postrema and the nucleus solitarius) involved in the vomiting reflex [3]. Although aprepitant treatment is well tolerated, there may occur some gastrointestinal track and CNS mild adverse effects. This NK1R antagonist, importantly, is an inhibitor and inductor of xenobiotic metabolism fundamental cytochromes CYP3A4 and CYP2C9, respectively [67,70]. It is worth to keep in mind that aprepitant may affect the plasma concentration of many chemotherapeutics and dexamethasone (or other narrow therapeutic margin drugs like warfarin).

Additionally, in the CINV prevention, other NK1R antagonists have been evaluated, but only three obtained market authorization [71,72,73,74,75]. Rolapitant (Varubi^®^ or Varuby^®^, Tesaro) received FDA and EMA approvals in human application, maropitant (Cerenia^®^, Zoetis and Prevomax^®^, Le Vet) received approval for the prevention of acute vomiting or vomiting due to motion sickness in veterinary clinics. Finally, netupitant, used in combination with the 5-HT_3_R antagonist palonosetron, also received approval for human use (Akynzeo^®^, Helsinn Birex). Casopitant (Rezonic^®^ or Zunrisa^®^, GlaxoSmithKline) nearly received approval, but GlaxoSmithKline withdrew its EMA authorization proposal following further safety assessments.

Many NK1R antagonists were explored as potentially therapeutic agents during early phase clinical trials [75,76,77,78,79]. A number of these were found ineffective or insignificantly useful in the treatment of major depressive disorder, depression, social anxiety, phobias, post-traumatic stress disorder, insomnia, schizophrenia, cannabis and opioid dependency, irritable bowel syndrome and overactive bladder, hot flashes, tinnitus and hearing loss, migraines, and painful diabetic polyneuropathy, and synergistically in antiviral HIV-1 therapy. Nevertheless, in a few studies of the NK1R antagonists, these compounds were revealed to be active in comparison to the placebo control. These successful reports refer to treatment of alcohol dependency (LY-686,017 case [80]) or chronic pruritus (aprepitant and serlopitant cases [81,82]).

On the other hand, great expectations are arising from the application of NK1R antagonists for the targeted treatment of malignant tumours. The antitumour action of NK1R antagonists (Table 3) was demonstrated in vitro on multiple human cell lines overexpressing NK1R as well as through in vivo research in xenograft murine models [2,16,17,83,84,85,86,87,88,89,90,91,92,93,94,95,96,97,98].

Recent studies refer to high specific antagonist nonpeptide agents, which efficiently compete with the SP/NK1R pathway. Despite structural differences of the presented antagonists (Figure 3), all cause direct receptor blockage correlated with a broad spectrum of anticancer effects. This pleiotropic activity may be highly valuable in terms of tumour treatment. Several NK1R ligands are also effective antiemetic agents. This is the case with aprepitant, the most studied NK1R antagonist with well-known pharmacokinetic and safety characteristics [2,15,39,66,67,68,69,78,99]. Alternatively, NK1R peptide ligand cyclosporine A, a commonly used immunosuppressive drug, also shows antitumour activity by binding to NK1R, as was demonstrated on various cancer cell lines [100,101].

Taken together, these reports suggest that NK1R antagonists may exert a wide range of potential therapeutic actions which may expand their clinical application. Despite the fact that there have been no oncological clinical trials executed to date to verify reported data, NK1R antagonists may prove to be a significant advance in the development of specific targeted antitumour therapy.

## 3. NK1R Radioligands in Nuclear Medicine

In nuclear medicine, ligands play the role of vectors leading medicine applied radionuclides to the target disease sites, which overexpress receptors for the implicated ligand. Radionuclide complexes are stable coordination compounds in which radionuclide cations are bonded by multidentate chelators (e.g., DOTA, NOTA, DOTAGA and DTPA). Syntheses of radioligands (radiopharmaceuticals) are performed according to the commonly known and described procedures, coupling reaction of ligand and bifunctional agent and labelling the obtained bioconjugate with the desired radionuclide.

### 3.1. Radiolabelled NK1R Agonists for Targeted Radionuclide Tumour Diagnosis

Diagnostic radiotracers (diagnostic radiopharmaceuticals) contain γ or β^+^-emitting radionuclides for SPECT and PET imaging, respectively. Radiopharmaceuticals are administered in such small amounts (at the nanomolar level) that they do not induce any pharmacological responses. After patient administration, the measurement of emitted gamma ray intensity allows for determination of the localization of the radiopharmaceutical and definition of tissue and organ abnormalities in the patient. These diagnostic methods enable detection of the biochemical and molecular pathologies at early disease stages, much earlier than the symptoms may be detected by the standard methods.

There are numerous studies describing the application of SP and its analogues or derivatives, labelled with diagnostic radionuclides (e.g., ^68^Ga, ^99m^Tc, ^111^In, ^125^I) designed for targeted radionuclide tumour diagnosis [102,103]. Table 4 presents concise information about published studies.

Nuclear characteristics of applied diagnostic radionuclides: ^3^H: emitter β^−^, t_1/2_ = 12.32 y, E_max_ = 18.59 keV; ^68^Ga: emitter β^+^, t_1/2_ = 67.71 min, E_mean_ = 0.836 MeV; ^99m^Tc: emitter γ, t_1/2_ = 6.01 h, E_max_ = 0.141 MeV; ^111^In: EC decay, emitter γ, t_1/2_ = 2.80 d, E_max_ = 0.245 and 0.171 MeV; ^125^I: EC decay, emitter γ, t_1/2_ = 59.41 d, E_max_ = 27.47, 27.20 and 35.49 keV.

In order to label SP with the iodine-125 radionuclide, Phe in position 8 was replaced by Tyr. The [Tyr^8^]SP was radioiodinated with ^125^I by the Bolton–Hunter agent (3-(3-iodo-4-hydroxyphenyl)propionic acid *N*-hydroxysuccinimide ester, [^125^I]I-BH). The experiments with radioligand [^125^I]I-BH-[Tyr^8^]SP showed specific, rapid and temperature-dependent binding of radiobiomolecules, as well as internalisation into pancreatic acinar cells derived from guinea pigs [104]. The same Bolton–Hunter iodinated SP derivative was used as a radiotracer for characterization of its binding via autoradiography and/or internalisation into various organs, tissues and cells [104,105,106,107,108,109,110,111,112,113,114,115,116,117,118,119,120] (Table 4).

The radioligand [^3^H]H-SP was examined in order to determine accumulation in feline urinary bladder interstitial cystitis (IC) [121]. The results showed low uptake of [^3^H]H-SP in normal and inflamed tissues, while high accumulation was discovered in inflamed bladder tissue and small blood vessels. In this case [^3^H]H-SP appeared to be specific only for inflamed bladder of cats diagnosed with IC, possibly due to upregulation of NK1R as a part of the IC pathophysiology.

Further, the endogenous peptide SP has been used for syntheses of numerous radioligands. Radiobioconjugate [^111^In]In-DTPA-[Arg^1^]SP was used for the in vivo detection of SP receptor-positive (SPR+) immunologic disorders and certain tumours [122]. The in vitro binding and autoradiographic experiments performed on parotid gland, brain cortex membranes and the submandibular gland of rat, demonstrated high affinity of [^111^In]In-DTPA-[Arg^1^]SP to NK1Rs. Tissue distribution of radioligand in male Wistar rats 24 h after treatment with 3 MBq of [^111^In]In-DTPA-[Arg^1^]SP, revealed high concentration of radioactivity in the kidneys indicating renal excretion as the central route of radiobioconjugate elimination. The experimental results showed also rapid enzymatic degradation of the tested radiocompound resulting in an approximately 3 min half-life in blood. Interestingly, no significant uptake of [^111^In]In-DTPA-[Arg^1^]-SP in the brain cortex and striatum was observed. These data suggested that the radiobioconjugate was unable to cross the intact blood–brain barrier (BBB) and further visualize SP receptors (SPRs) in the central nervous system. Additionally, [^111^In]In-DTPA-[Arg^1^]-SP injected into rats bearing the autograft pancreatic tumour, CA20948, or rats with adjuvant mycobacteria tuberculosis-induced arthritic joints, exhibited significant uptake in the tumour, salivary glands, kidneys and arthritic hind leg joints. The authors further demonstrated the potential of this radiolabelled SP analogues for visualisation of pathological SPR+ processes in vivo by gamma camera scintigraphy.

The same [^111^In]In-DTPA-[Arg^1^]SP radiobioconjugate was involved in the first clinical trials on twelve patients with immune-mediated diseases in 1996 [123]. After intravenous administration of 150-250 MBq of [^111^In]In-DTPA-[Arg^1^]SP, rapid radiopharmaceutical degradation, within four minutes of treatment, was observed. Twenty-four hours postinjection, more than 95% of radioactivity was excreted in the urine. The uptake in areolae mammae (in women), liver, spleen, kidneys and urinary bladder was observed in all patients and in the thymus in eight patients. This radiobioconjugate can be used for scintigraphy of inflammatory sites in various diseases as well as for visualisation the thymus.

Studies of the diagnostic properties of ^99m^Tc radiolabelled IMB-SP bioconjugate [124] have revealed uptake in the salivary glands, while the accumulation was decreased by factor of 2 in mice pretreated with excess of non-radiolabelled SP. The authors did not observe any cardiovascular side effects due to the slow rate of SP infusion.

SP ligand conjugated with Hynic chelator or monodentate bifunctional chelator isocyanobutyric succinimidyl ester and labelled with technetium-99m, [^99m^Tc]Tc-Hynic-SP, [^99m^Tc](NS_3_)-Tc-CN-SP and [^99m^Tc]((NS_3_)-Tc-CN)_2_-SP have been synthesised and studied to determine potential application in targeted radionuclide tumour diagnosis [125]. The radiobioconjugates were characterized by high stability in the presence of competitive cysteine/histidine solutions and various lipophilicity (logP) values, of −3.7, −0.24 and −0.89, respectively. SP ligands conjugated with two cysteine amino acids (Cys-Cys-SP) were used in the studies to apply ^99m^Tc and ^188^Re as theragnostic matching pairs based on the combined reaction of tridentate π-donor and monodentate π-acceptor chelators with the [Tc/Re≡N]^2+^ metallic functional group [126]. [^99m^Tc][Tc(N)(Cys-Cys-SP)(PCN)] and [^188^Re][Re(N)(Cys-Cys-SP)(PCN)] radiobioconjugates incubated with U87MG cells expressing NK1R displayed predominant cell surface binding, whereas incubation with negative control cell line, L-929, resulted in no detectable interaction. Whole-body biodistribution studies using hybrid SPECT/CT YAP(S)PET small-animal tomography showed significant kidney and thymus uptake, in accordance with previous studies [123]. High radioactivity accumulation in salivary glands was also detected. The presented results ruled out myocardial uptake but did not confirm if the uptake of synthesised bioconjugates was specific.

The SP analogue [Tyr^8^,Met(O_2_)^11^]SP conjugated with DOTA chelator and labelled with diagnostic radionuclides ^111^In or ^68^Ga was used in glioma patient studies. The [^111^In]In-DOTA-[Thi^8^,Met(O_2_)^11^]SP [55] and [^68^Ga]Ga-DOTA-[Thi^8^,Met(O_2_)^11^]SP [127] receptor radiopharmaceuticals were injected simultaneously with therapeutic preparation [^213^Bi]Bi-DOTA-[Thi^8^,Met(O_2_)^11^]SP to visualise distribution of radiopharmaceuticals in the whole body.

Similar to previous studies, SP analogue [Tyr^8^,Met(O)^11^]SP was applied for the synthesis of bioconjugate Hynic-[Tyr^8^,Met(O)^11^]SP. The obtained bioconjugate, labelled with ^99m^Tc, was used for detection of NK1R positive tumours [128]. The obtained radiobioconjugate was characterized with high specific activity (84–112 GBq/μmol) and stability in human serum. Internalisation studies on U373 MG cells, an astrocytoma cell line, showed rapid (after 0.5 h) binding of the tested compound to the cell membrane and specific internalisation. Saturation binding assays indicated a mean K_d_ value in the nanomolar range, confirming the radioligand specificity to NK1R. Biodistribution studies in healthy mice and in tumour bearing nude mice demonstrated specific uptake in the tumour and the noticeable uptake in the stomachs, intestines and lungs, as well high accumulation in the kidneys, indicating excretion via the renal pathway. There was not significant accumulation in the salivary glands although this accumulation was moderately higher than in the other organs, including muscles and bones.

SP analogues [Pro^9^]SP and [Met(O_2_)^11^]SP(7–11) labelled with ^3^H radionuclide were applied in studies of different NK1R binding sites [11]. The researchers examined internalisation of two radioligands, [^3^H]H-[Pro^9^]SP and [^3^H]H-propionyl[Met(O_2_)^11^]SP(7–11), using CHO cells transfected with human NK1Rs. The results showed existence of two nonstoichiometric binding sites NK-1M (majority) and NK-1m (minority). Both radiobioconjugates were internalized rapidly to achieve a maximum of 75% for specifically bound [^3^H]H-[Pro^9^]SP and of 35% for [^3^H]H-propionyl[Met(O_2_)^11^]SP(7–11). [^3^H]H-[Pro^9^]SP interacted with the most abundant NK-1M binding site inducing adenylyl cyclase activation (temperature dependent internalisation), whereas [^3^H]H-propionyl[Met(O_2_)^11^]SP(7–11) bound to the less abundant NK-1m binding site connected with the phospholipase C (PLC) pathway (temperature independent internalisation).

The binding properties of the [^125^I]I-BH-[Sar^9^,Met(O_2_)^11^]SP radiocompound based on the ligand [Sar^9^,Met(O_2_)^11^]SP were tested using autoradiography and compared with properties of [^125^I]I-BH-SP [129]. The results showed high uptake of both radiobiomolecules in the submandibular gland and in several regions of rat brain. Parallel in vitro binding experiments on rat brain membranes exhibited two to four fold higher affinity of [^125^I]I-BH-[Sar^9^,Met(O_2_)^11^]SP than that of [^125^I]I-BH-SP. Despite higher affinity, [^125^I]I-BH-[Sar^9^,Met(O_2_)^11^]SP did not demonstrate significantly higher specificity than [^125^I]I-BH-SP in NK1R binding sites, in agreement with previous studies of these two radiobiomolecules [130]. However, autoradiographic data of [^3^H]H-[Sar^9^,Met(O_2_)^11^]SP reported in [131] showed slightly different localization of NK1Rs in the rat brain compared to previously presented data for [^125^I]I-BH-SP [130].

### 3.2. Radiolabelled NK1R Agonists for Targeted Radionuclide Tumour Therapy

Therapeutic radiopharmaceuticals used in cancer treatment or palliative therapy are often ligands labelled with radionuclides which emit corpuscular radiation of short range in the tissue (e.g., β^−^, α or Auger electrons). Targeted radiopharmaceuticals characterized by high receptor affinity are selectively absorbed in the pathological tissues and their radiation energy is selectively and quantitatively deposited in the tumour mass. As a result, applications of these preparations are relatively safe for healthy tissues.

Nuclear characteristics of applied therapeutic radionuclides [102,103]: ^90^Y: emitter β^−^, t_1/2_ = 64.00 h, E_max_ = 2.28 MeV, mean tissue range: 2.76 mm; ^177^Lu: emitter β^−^, t_1/2_ = 6.65 days, E_max_ = 0.50 MeV, max. tissue range: 0.28 mm; ^213^Bi: emitter α, t_1/2_ = 45.59 min, E_max_ = 5.88 and 5.56 MeV, tissue range: 40–100 μm; ^225^Ac: emitter α, t_1/2_ = 9.92 d, E_max_ = 5.83 and 5.80 MeV, tissue range: 40–100 μm.

In addition to the high incidence of NK1Rs in GBM, NK1Rs are overexpressed in approximately 27% of human pancreatic tumours [132]. Bortoleti de Araújo et al. applied endogenous SP for the synthesis of [^177^Lu]Lu-DOTA-SP radiobioconjugate (Table 5) and evaluated in vivo targeting of AR42J pancreatic tumour cells in Nude mice [133]. This study, along with others, reported that the radiobioconjugate was stable for more than 24 h at 37 °C in human plasma. Biodistribution studies on AR42J pancreatic tumour bearing mice showed high uptake in kidneys, suggesting excretion mainly by renal pathway. Significant uptake of [^177^Lu]Lu-DOTA-SP was also observed in intestine and stomach due to the presence of NK1Rs in the gastrointestinal tract. These results demonstrated the potential of [^177^Lu]Lu-DOTA-SP as a treatment for pancreatic tumours.

Considering the high expression of NK1R on malignant glial brain tumours [132], studies of SP ligands have been initiated. The first in vivo studies concerned application of a SP with macrocyclic DOTAGA chelator, labelled with therapeutic radionuclides (mostly ^90^Y and to reduce the “cross-fire effect”, ^177^Lu and ^213^Bi), were performed by Kneifel et al. [134]. In clinical experiments with twenty patients, the radiopharmaceutical was administered via an implanted catheter directly into the tumour mass or via intracavitary implant after surgical resection (the local injection minimizes side effects and reduces the tubular reabsorption of the radiopharmaceutical in the kidneys). Malignant glioma therapy of WHO grade 2 to 4 tumours, by applying β^−^ and α emitters, was well tolerated with low toxicity and resulted in the radiation induced necrosis of cancer cells. However, due to infiltrative characteristics of GBM, its complete surgical resection cannot be achieved. Therefore, the novel approach of neoadjuvant therapy, with the use of ^90^Y radionuclide as the primary therapeutic modality, was proposed by Cordier et al. [135]. Seventeen patients with newly diagnosed and histopathologically confirmed GBM were treated with [^90^Y]Y-DOTAGA-SP before tumour surgical resection. The catheter systems were stereotactically implemented within the cancer margins and [^90^Y]Y-DOTAGA-SP radiopharmaceutical (radioactivity ranged from 120 mCi to 345 mCi) was injected intratumorally. During the treatment, no increase or decompensation of intracranial pressure was observed. The pretreated tumours were demarcated by a capsule structure, leading to better separation from the cerebral tissue than in conventional glioma resection. Moreover, pseudo-encapsulation allowed for a marked reduction of intraoperative bleeding. The highest dose administered in ten patients caused the completed encapsulation of the tumour. This neoadjuvant local therapy was feasible without significant side effects, 15 of 16 patients treated so far exhibit stabilisation of neurological status. However, in critically located gliomas the use of ^90^Y radionuclide with the resulting “cross-fire effect” may cause unacceptable damage of adjacent brain areas.

To minimize the neurological damage, the alpha radiation-emitting radionuclide ^213^Bi, which has a shorter range in tissue and higher radiation energy in comparison with ^90^Y and ^177^Lu, was the first-line treatment [55]. Moreover, for the synthesis of novel radiopharmaceuticals, the [Thi^8^,Met(O_2_)^11^]SP ligand, characterized by a longer half-life in vivo, was applied [53]. This SP analogue has been selected from various SP analogues tested (Table 2) in terms of the feasibility of vector application usage in targeted radionuclide tumour diagnosis or therapy. Pilot studies with [^213^Bi]Bi-DOTA-[Thi^8^,Met(O_2_)^11^]SP and [^111^In]In-DOTA-[Thi^8^,Met(O_2_)^11^]SP were conducted in five patients diagnosed with critically located gliomas (WHO, grade 2-4) [55]. Similarly to previous clinical trials, the radioactive compound [^213^Bi]Bi-DOTA-[Thi^8^,Met(O_2_)^11^]SP was administered intratumorally via implanted catheters. Simultaneously, the [^111^In]In-DOTA-[Thi^8^,Met(O_2_)^11^]SP [55] or [^68^Ga]Ga-DOTA-[Thi^8^,Met(O_2_)^11^]SP [127] radiocompounds were used for the visualisation of ^213^Bi-radiopharmaceutical distribution. Sixty to ninety minutes postinjection of [^213^Bi]Bi-DOTA-[Thi^8^,Met(O_2_)^11^]SP, less than 4% of the activity was present in the blood, confirming high retention of this radiopharmaceutical in target site. The preliminary studies showed that this method is well tolerated and safe for patients; however, the short half-life of ^213^Bi reduced the effectiveness of the treatment of larger tumours. Based on statistical data presented in the literature [127,136,137,138], the local treatment of brain tumours (intracavitary administration of radiopharmaceuticals) with the use of NK1R ligands in patients suffering from secondary GBM compares favourably with standard treatment options.

[Thi^8^,Met(O_2_)^11^]SP ligand was also used for in vitro studies of potential ^225^Ac-radiopharmaceuticals for targeted therapy of NK1R expressing gliomas performed by Majkowska et al. [139]. The efficacy of [^225^Ac]Ac-DOTA-[Thi^8^,Met(O_2_)^11^]SP was tested on three human glioblastoma cell lines (T98G, U87MG, U138MG), as well as GSCs. The binding experiments performed on T98G cells demonstrated high affinity (K_d_ = 19.2 ± 1.9 nM) of the radiobioconjugate for NK1R, which agrees with previous studies [140]. [^225^Ac]Ac-DOTA-[Thi^8^,Met(O_2_)^11^]SP caused significant reduction in glioblastoma cell viability compared to the conventional treatment with chemotherapeutic temozolomide. This radiobioconjugate has been shown to induce apoptosis and cell cycle arrest in G2/M phase. Importantly, [^225^Ac]Ac-DOTA-[Thi^8^,Met(O_2_)^11^]SP was found to be highly cytotoxic, not only towards established GBM cell lines, but also to GSCs cells which are particularly resistant to radio- and chemotherapy. According to the literature [141,142,143], these stem cells (variable from 1% or less in the case of low-grade tumours and to 30% in highly aggressive glioblastomas) are responsible for initiation of tumour formation in vivo, sustaining tumour growth, and contributing to the creation of metastatic lesions. Therefore, therapy which can target GSCs and GBM cells, such as [^225^Ac]Ac-DOTA-[Thi^8^,Met(O_2_)^11^]SP therapy, may decrease the recurrence of gliomas and improve survival rate. Clinical trials using this radiobioconjugate have been initiated [144]. 

However, the main disadvantage of using [Thi^8^,Met(O_2_)^11^]SP ligand as a vector in ^213^Bi/^225^Ac-radiopharmaceuticals is poor radiopharmaceutical diffusion into the walls of the postsurgical cavity. Following these results, fragments of SP ligand and its derivatives (SP(5–11), SP(4–11), [Thi^8^,Met(O_2_)^11^]SP(5–11)) were tested as potential vectors to guide the radiobioconjugate to the NK1Rs expressed on cancer cells [145]. These studies were focused on the synthesis of new radiobioconjugates with higher lipophilicity and lower molecular weight than those of [Thi^8^,Met(O_2_)^11^]SP. The new radiobioconjugates were projected to be more effective at diffusing into solid tumours or postsurgical cavity walls. The results of these studies showed that shorter fragments of SP were characterized by a lower molecular weight and higher lipophilicity, whereas the exchange of Phe for Thi in position 8 and Met for MetO_2_ in position 11 caused a decrease of lipophilicity. The newly prepared radiobioconjugates, such as [^177^Lu]Lu-DOTA-SP(4–11) and [^177^Lu]Lu-DOTA-[Thi^8^,Met(O_2_)^11^]SP(5–11) exhibited satisfactory affinity to NK1Rs presented on U373 MG glioblastoma cells with K_d_ values in the nanomolar range. Unfortunately, radiolabelled shorter fragments of SP in HS were less stable than in CSF due to fast enzymatic ligand biodegradation. Considering the lipophilicity, molecular weight, affinity and stability of SP, its fragments and analogues, a balance should be feasible. The relatively low ligand stability in HS does not completely disqualify the local treatment of radiolabelled short SP fragments, because the content of serum peptidases in the cavity after tumour resection is rather low. Nevertheless, the application of receptor radiopharmaceuticals requires individual consideration (personalised medicine) for each patient.

The above paragraphs of this review cover rich aspects of the biological properties and potential applications of NK1R agonist radioligands in targeted radionuclide diagnosis and therapy. The cited literature are quite consistent besides stability evidence of radiobioconjugates ^99m^Tc-Hynic-[Tyr^8^-Met(O)^11^]SP [128] and [^177^Lu]Lu-DOTA-SP [133]. In contrast to plenteous literature, only these two papers provide data about radioligand stability in human serum. In our opinion this is questionable, due to endopeptidases present in HS leading to inevitable process of SP biodegradation.

Based on the data described in Section 3.1 and Section 3.2, it is possible to conclude that NK1R ligands, labelled with diagnostic/therapeutic radionuclides, can be applied for imaging or therapy of cancers overexpressing the NK1Rs and for that these receptor radiopharmaceuticals are efficacious. This therapy seems to be prospective for the future in comparison to standard applied therapeutic options like temozolomide chemotherapy. Due to the relatively low stability of SP, its fragments and analogues, new NK1R ligands are constantly being investigated, starting innovative studies with the application of SP antagonists.

### 3.3. Antagonist Radioligands of NK1R for Targeted Radionuclide Imaging

A number of radiolabelled antagonists for NK1R are known to date. All radioligands exhibit high affinity to NK1R, but only some have potential future applications in targeted radionuclide tumour diagnosis or therapy. The features of those with potential are presented below (Table 6).

In contrast to SP and its peptide analogues, nonpeptide antagonists are stable and do not follow rapid enzymatic decomposition in vivo. First generation radiotracers were selected from many lead structures, based upon autoradiographic studies of NK1Rs. These compounds were initially investigated in vivo for their ability to be taken up by the brain and other organs in animal models [146,147,148,149] (Figure 4). Early evaluation of [^11^C]CP-96,345 [146] concluded that despite radioligand high lipophilicity, the cationic form excluded radiotracer passage through the blood–brain barrier and allowed for peripheral tissue imaging only. PET detection of central and peripheral NK1R occupancy was performed later on hamsters using [^11^C]CP-99,994 [147] and pigs using [^11^C]CP-643,051 [148]. Both studies confirmed the known autoradiographic mappings. The first primate PET imaging was reported in 2000 at the Uppsala University PET Centre [149], and revealed the uptake kinetics of [^11^C]GR205171 ([^11^C]vofopitant) (Figure 5). Research includes regular imaging (baseline study) and imaging with receptor blocked active sites (using unlabelled compound) to demonstrate high specific binding in the striatum, thalamus and neocortex. Tracer analysis showed rapid uptake during 50 min of examination and fast cerebellar nonspecific washout, which introduced the possibility of in vivo assessment of receptor distribution. At the same time, the slow dissociation rate of this tracer from NK1Rs excludes dynamic PET imaging using [^11^C]GR205171 for physiological receptor regulations. Thereby, the tracer evaluations in human were focused on brain NK1R mapping for further potential clinical application.

Alternative research were performed using similar highly specific radiotracer [^18^F]SPA-RQ, a ^18^F-labelled derivative of vofopitant (Figure 5), US patented by Merck & Co. in 2001 and investigated mainly in Merck Research Laboratories (West Point, Pennsylvania) and Turku Positron Emission Tomography Centre. The influence of gender and age on NK1R availability was investigated using [^18^F]SPA-RQ 3D PET imaging [150], including 35 male and 10 female volunteers, aged 19 to 55 years. Results revealed a significant relationship of general 7% decrease rate of cerebral NK1Rs per decade of life, caused by physiological aging. This phenomenon was observed in frontal, temporal, parietal cortex and hippocampus structures. Moreover, NK1R availability, especially in the striatum, was found to be relatively lower in women than in men, what was later confirmed using [^11^C]GR205171. Indeed, further investigations [151] provided data of similar interaction effects of age and sex on cerebral receptor availability using this radiotracer. These results indicated similar declines in the rate of NK1 receptor density in the frontal, temporal, and occipital cortices, but also in the brainstem, thalamus and caudate nucleus per each decade of life. Surprisingly, receptor availability loss in the amygdala and temporal cortex with an age increase was noted only in men. Additionally, in this study women showed lower general NK1R density in the thalamus compared to male volunteers.

[^18^F]SPA-RQ and [^11^C]GR205171 are based on the same pharmacophore. Both demonstrate fast brain uptake and very high affinity for NK1R with low nonspecific binding. Both radioligands enable efficient parametric PET imaging via a simple method based on reference ratio of region of interest (ROI) signal to cerebellar signal. This is a capability of the radioligands referenced due to fast washout of nonspecific bond tracer in the cerebellum bereft of NK1Rs. All of these features favour application of both radiotracers in localization and quantification of receptor studies in course of targeted radionuclide therapy.

Preclinical evaluation of PET NK1R imaging in human brains, using [^18^F]SPA-RQ, was performed in terms of further investigations of NK1R antagonists, accurate receptor neurodistribution and influence of receptor regulation on CNS pathologies. PET 3D studies [152] detailed multiple observations including the highest uptake of the radiotracer at the putamen and caudate, followed by uptake rates in adjacent substantia nigra and globus pallidus structures. Uniform and moderate uptake of radiotracer was widespread in limbic cortex and neocortex regions, while inconsiderable specific uptake was observed in the cerebellum. Autoradiographic post-mortem studies on the human brain confirmed similar affinity. However, examination via kinetic modelling revealed a few drawbacks among visualisation procedure. First, the low rate of radiotracer kinetics required a six hour time period for the binding equilibrium to occur (Figure 6). This time impacted imaging quality and reliability, but also favoured the radioactive agent metabolic decomposition; 90 min after injection, only 40% of the radioactivity in plasma still represented the initial [^18^F]SPA-RQ concentration. Moreover, free fluoride was observed, likely resulting in the skull bone radioactivity that was observed during late scans In the end, a simplified reference tissue kinetic model of the radiotracer uptake was optimized. After 240 min, data acquisition was optimal, corresponding to set binding potential (BP, a combined measure of the density of “available” neuroreceptors and the affinity of a drug to this neuroreceptor). BP values were assigned from 4 to 5 in the basal ganglia structures and between 1.5 and 2.5 in the cortical regions, in reference to the cerebellum. Researchers also suggested that for clinical imaging utility, a commonly applied ratio method considering cerebellar reference is reliable and preferable, as it does not require arterial blood sampling and long acquisition time.

In support of these measurements, a similar radioligand [^18^F]fluoroethyl-SPA-RQ ([^18^F]FE-SPA-RQ), with high affinity and selectivity to NK1R was developed and examined [153,154]. [^18^F]FE-SPA-RQ featured lower in vivo defluorination relative to the previous analogue, [^18^F]SPA-RQ, seen via reduced radioactivity in bone accumulation. A group from the Japanese Molecular Imaging Centre (National Institute of Radiological Sciences, Chiba, Japan) examined [^18^F]FE-SPA-RQ characteristics on primates and rodents. These studies were followed by PET imaging with kinetic modelling to validate visualisation and quantification of NK1R distribution in healthy men. Although similar [^18^F]SPA-RQ and [^18^F]FE-SPA-RQ imaging of distribution on human brain receptors was observed; BP values obtained for [^18^F]FE-SPA-RQ were lower, in fact, [^18^F]FE-SPA-RQ showed little higher affinity for NK1R than [^18^F]SPA-RQ.

All of these explorations enable further investigations of NK1R system contribution in targeted radionuclide therapy of CNS pathologies. So far, several early phase clinical trials using the discussed radiotracers have been conducted. Temporal lobe epilepsy [155] and social anxiety disorder [156] PET studies using [^11^C]GR205171 were performed in small studies comparing NK1R availability in specific cerebral structures of patients and healthy control subjects. In both studies, applied radiotracer imaging provided positive correlations between increased receptor availability and patient neuropathology. More precisely, significant enhanced in NK1R availability was determined in the ipsilateral and contralateral hemispheres of the temporal lobe in epileptic patients and in the right amygdala in patients with social anxiety disorder (Figure 7). Moreover, increased receptor availability correlated with epileptic seizure frequency in ipsilateral medial temporal structures. Another [^11^C]GR205171 imaging study was performed on healthy volunteers with snake or spider phobias to investigate the effect of fear stimuli on NK1R availability [157]. During PET acquisitions, subjects were provoked by normal and fear stimulating animal pictures to distinguish baseline and fear stimulated scans of NK1R occupancy imaging. As a result, provocation of specific phobia had influence on signals in the right amygdala, which suggested that fear stimulated the release of endogenous SP in that region of the brain. Taken together, these findings demonstrate that [^11^C]GR205171 voxel-based statistical analysis can substantially support NK1R mapping in specific anxiety, stress and epileptic disorders.

[^18^F]SPA-RQ was also employed in the quantification of NK1Rs in the brains of patients with panic disorder [158,159]. Baseline PET imaging and imaging after pharmacological induction of panic attacks were performed in patients and healthy volunteer cohorts. The result revealed a significant reduction of NK1Rs at baseline in patients compared with healthy subjects in all brain ROIs (12% to 21% reduction). Although this is in contrast to the above study focused on patient with social anxiety disorder, a similar outcome was demonstrated when two groups of female subjects were compared, with and without Irritable Bowel Syndrome [160]. Based on PET imaging supported with MRI scans, comparison of cerebral NK1R density revealed lower [^18^F]SPA-RQ binding levels in patients in the globus pallidus. This radiotracer was also used in trials focused on post-traumatic stress disorder [161] and effective PET quantification of pancreatic cancer lesions for targeted radionuclide tumour therapy [162].

Meanwhile, it should also be noted that radiotracer molecular imaging is increasingly being used to determine receptor pharmacodynamics and optimize applicable effective doses of novel NK1R antagonists. One of the first clinical trials focused on this inspected a highly selective antagonist, aprepitant, while it was undergoing market authorisation. Determination of relationship between brain NK1R occupancy levels and aprepitant oral dose or plasma concentration was explored using [^18^F]SPA-RQ in single-blind, randomised, placebo-controlled studies in healthy participants [163]. Indeed, reliable cerebral NK1R occupancy prediction, according to aprepitant plasma concentration was determined. More than 90% of striatal receptor occupancy was reached in subjects with aprepitant plasma concentration around 100 ng/mL, after >100 mg/day of aprepitant for 2 weeks. Even though the study was conducted with a limited number of subjects and small variety of doses, it provided valuable indications for dosage regimens by which to achieve effective receptor blockage in the CNS (Figure 8).

NK1R occupancy PET mapping by the [^18^F]SPA-RQ radiotracer was also applied in later clinical trials as verification of adequate target drug dose engagement. NK1R antagonists, aprepitant and its analogue L-759,274, were investigated in trials treating participants with major depressive disorder and anxiety disorder, respectively [76,78]. Unfortunately, both double-blind, randomised, placebo- and active-controlled, multicentre phase III clinical trials indicated not effective activity of aprepitant and L-759,274 for the treatment of depression or anxiety disorders, respectively. However, these researches demonstrated that radiotracer imaging can be used as a supportive tool in the proper context to validate negatively terminated trials.

[^11^C]GR205171 tracer application in clinical trials was explored intensively by GlaxoSmithKlein. It is perfectly illustrated in determination of casopitant plasma concentration correlation with receptor occupancy [164]. Baseline and post-casopitant oral dose PET-[^11^C]GR205171 brain imaging was performed in addition to collection of subject plasma for pharmacokinetic characterizations. The main goal of this study was to develop an estimation of suitable casopitant dose for subsequent patient clinical trials. Only the 15–30 mg drug dose was shown to achieve NK1R occupancy above 95%, which is a higher concentration than previously reported casopitant doses applied in depression trials [165,166]. This quantitative result was based on radiotracer pharmacokinetic model of NK1R binding initially established in the study [165]. Similar PET studies using [^11^C]GR205171 were performed under phase 1 of interventional clinical trials to investigate receptor occupancy, safety, tolerability and pharmacokinetic features of rolapitant [167] and two other promising antagonists GSK1144814 [168] and GSK206136 [169]. Rolapitant PET evaluation indicated that a single oral treatment at 200 mg was a sufficient dose to occupy more than 90% of cerebral NK1Rs. Moreover, this receptor occupancy level was maintained for 120 h post-drug administration. These results suggest that treatment can provide properly long antiemetic action in case of highly emetogenic chemotherapy.

In 2005 at Amsterdam VU University Medical Centre a novel selective radiotracer [^11^C]R116301 was developed (Figure 9) [170]. This radiotracer was evaluated in potential PET visualisation of NK1R in the human brain [171,172]. Similarly, as in above demonstrated cases, the highest uptake was recorded at striatum and lower in thalamus and other cortex regions. The cerebellar uptake level was negligible and nonspecific, as is presented in previous NK1R distribution reports. Receptor blocking with 125 mg oral dose of aprepitant reduced significantly in the striatum and cortex imaging signals relative to the cerebellar level. Further investigation of [^11^C]R116301 specific binding was performed in 11 healthy volunteers using a test-retest method where baseline imaging and imaging after treatment with known doses of aprepitant were obtained in the same subjects. Researchers recommended to examine the ROI of the cerebellum as a semiquantitative ratio method for clinical applications, mainly due to [^11^C]R116301 fast specific binding kinetics enabling realization of PET scans in 60–90 min after injection. In comparison, [^18^F]SPA-RQ has slower kinetics, reaching cerebral receptor binding equilibrium only 6 h after treatment. Such slow kinetics present a considerable limitation to routine application, even if the given radiotracer provides relatively high specific imaging signal. This stands in accordance with NK1R-ligand affinity order [149], where the highest affinity ligands are GR205171 and SPA-RQ, followed by aprepitant, R116301 and SP. However, the use of [^11^C]R116301 in dynamic brain receptor imaging seems to be promising in various concentrations of endogenous SP, for instance, in pharmacological test investigations.

In the course of drug development and final market approval, radiolabelled aprepitant, casopitant and netupitant need to be studied in terms of bioavailability [173,174,175,176]. Determination of absorption, distribution, metabolism and excretion features were performed on ^14^C-labelled derivatives of listed antagonists, synthesised for a biotransformation evaluation in rats, dogs and humans. The use of long-lived radionuclide traces for that purpose enabled investigation a range of metabolites (excreted or blood circuiting) after intravenous or oral administration.

## 4. Conculsions

This review discusses precisely literature data concerning the chemical and biological aspects of natural and synthetic NK1R ligands in classical (targeted therapy) and nuclear (targeted radionuclide therapy) medicine. Despite that standard nonspecific cancer treatments (e.g. chemotherapy, radiation therapy) are still the dominant form of therapy, the specific cancer treatments - targeted radionuclide tumour therapy is increasingly used in clinical trials. Due to expression of NK1Rs in a wide variety of cancers, the NK1R/SP system can be used as a modulator of biological functions related to tumour cell proliferation (favouring tumour growth), angiogenesis and migration. However, because of relatively low stability of SP, its fragments and analogues, new NK1R ligands are constantly being investigated. It initiated innovative studies with the application of SP antagonists.

To summarize, use of PET radiotracers supports progress in NK1R antagonist development and facilitates clinical investigations in terms of NK1R system correlation with CNS pathologies. Moreover, molecular imaging has the potential to improve therapeutic monitoring and to explore the duration of pharmacological effects in course of targeted radionuclide therapy. Although several clinical trials using radiotracers based on NK1R antagonists have been conducted to date, no serious attempts have been made in the oncological space yet.

## Figures and Tables

**Figure 1 pharmaceutics-11-00443-f001:**
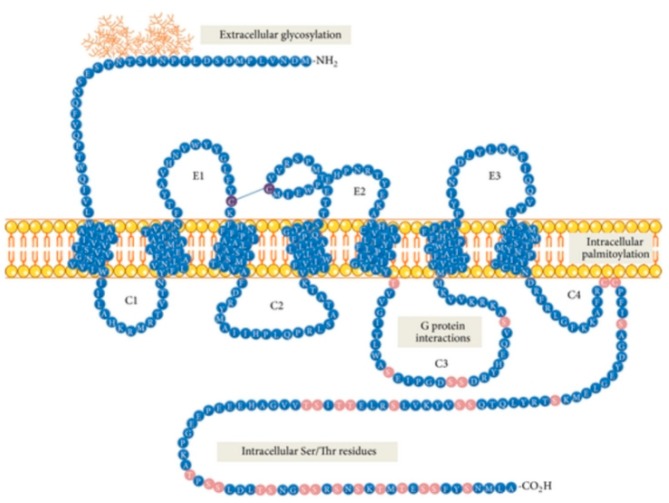
Schematic model of NK1 receptor of full length isoform (NK1R-Fl) [3].

**Figure 2 pharmaceutics-11-00443-f002:**
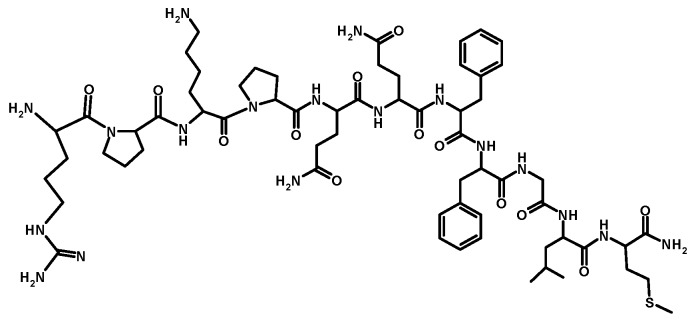
Structure of Substance P(1–11).

**Figure 3 pharmaceutics-11-00443-f003:**
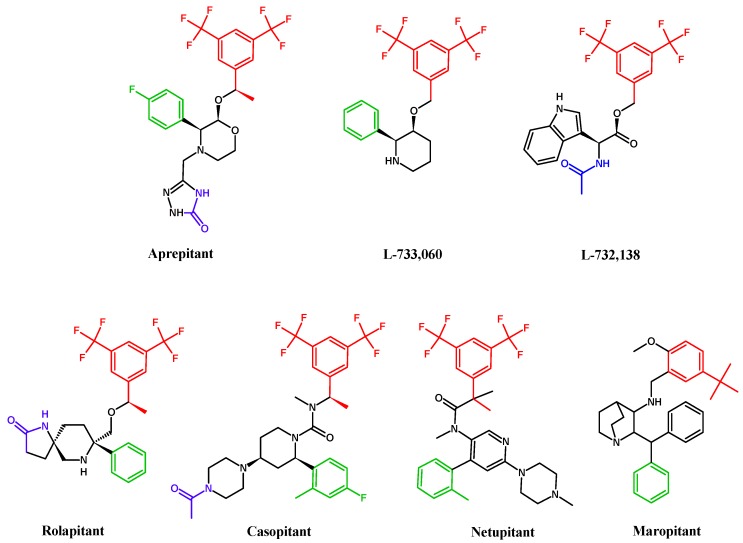
Colour visualisation of structure differences and similarities in the group of most antitumour evaluated NK1R antagonists (aprepitant, L-733,060 and L-732,138) and in group of CINV indicated NK1R antagonists (aprepitant, rolapinant, casopitant, netupitant and maropitant); the last presented, maropitant, despite the biggest structural distinction, showed high affinity to NK1R and sufficient clinical efficacy.

**Figure 4 pharmaceutics-11-00443-f004:**
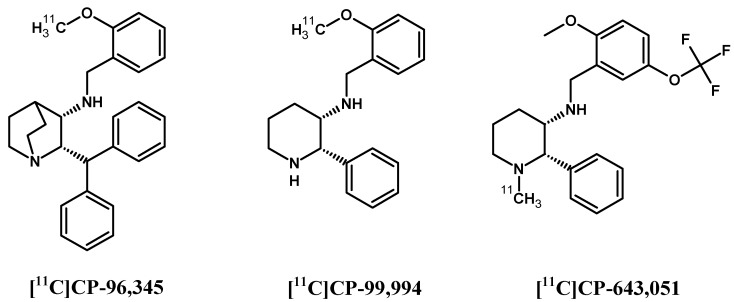
Structures of the first radioligands of neurokinin 1 receptor.

**Figure 5 pharmaceutics-11-00443-f005:**
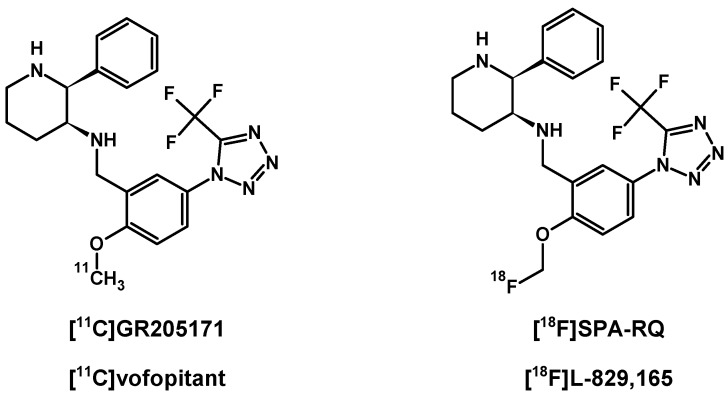
Twin structures of radiotracers [^11^C]GR205171 and [^18^F]SPA-RQ.

**Figure 6 pharmaceutics-11-00443-f006:**
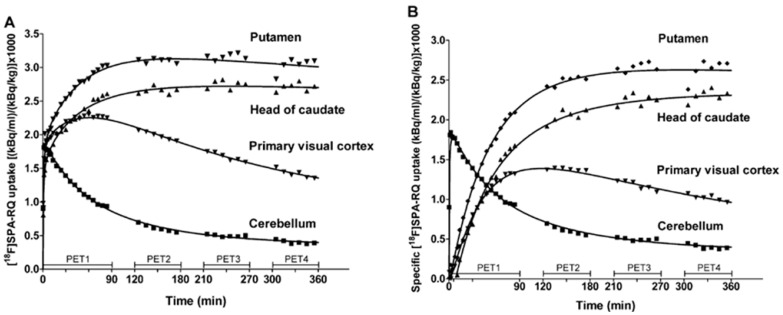
[^18^F]SPA-RQ radiotracer total (**A**) and specific (**B**) uptake values in function of time in striatal structures and occipital cortex. As reference, both charts present cerebellar total binding curves [152].

**Figure 7 pharmaceutics-11-00443-f007:**
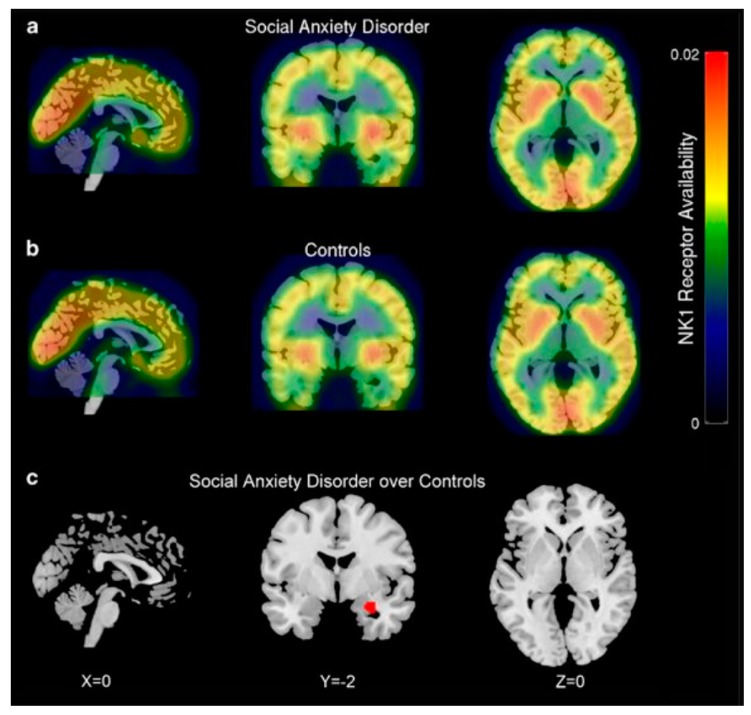
[^11^C]GR205171 PET/MRI imaging of NK1R occupancy in social anxiety disorder patients (**a**) and healthy controls (**b**). Mean PET parametric scans (in colour) of radiotracer were overlaid on MRI images and compared between both groups. Patients with social anxiety disorder showed increased NK1 receptor availability in the right amygdala (**c**) [156].

**Figure 8 pharmaceutics-11-00443-f008:**
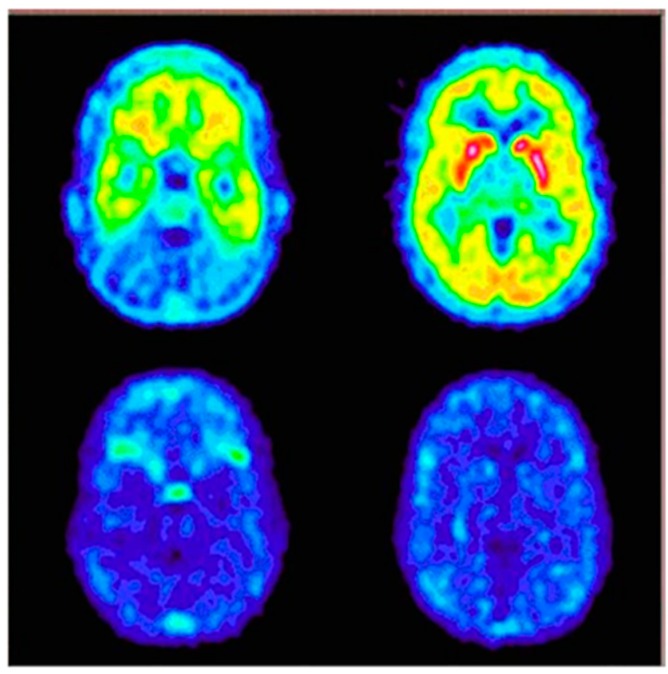
Predose (upper) and 100 mg aprepitant postdose (bottom) PET scans in the transverse section at the level of cerebellum (left) and striatum (right) in a human subject. Warmer colours symbolise higher uptake of radiotracer [^18^F]SPA-RQ. Based on ratio method with cerebellum reference, estimated receptor occupancy by aprepitant for this subject was 94% at 1053 ng/mL of aprepitant in plasma [163].

**Figure 9 pharmaceutics-11-00443-f009:**
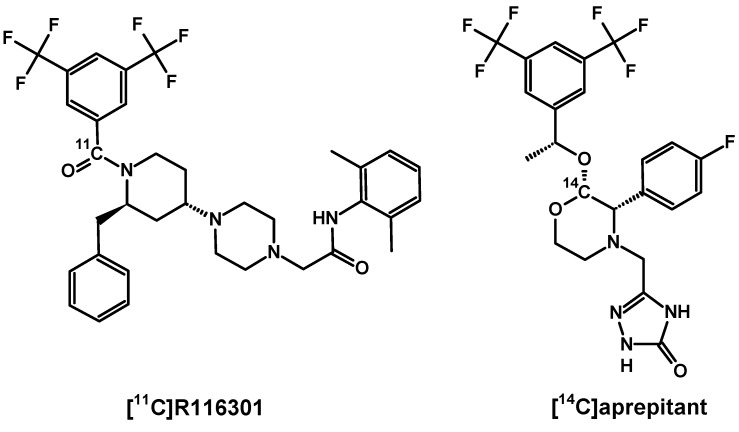
[^11^C]R116301 (left) and exemplary [^14^C]aprepitant (right) structures.

**Table 1 pharmaceutics-11-00443-t001:** Substance P, its analogues and derivatives: biological activity and potential application in classical medicine.

NK1R ligands	Ligand Biological Properties and Applications	References
**Mammalian NK1R ligands**		
	• Phosphatidylinositol signal pathway activation and intracellular calcium concentration increase;	[20]
	• Treatment of depression and associated anxiety;	[38]
	• Prevention of vomiting after anaesthesia or chemotherapy;	[39,40]
**Substance P** (SP, SP(1–11), [Arg^1^]SP)	• Increase of endothelial ion transport and permeability of vessels in tissue inflammation states;	[41,42,43,44]
	• Neuropathic pain modulation;	[45]
	• Liver cirrhosis biomarker;	[46,47,48]
	• Bone tissue metabolism modulator, especially of osteoblast activity at a later stage of bone formation;	[47,48]
Arg^1^-Pro^2^-Lys^3^-Pro^4^-Gln^5^-Gln^6^- Phe^7^-Phe^8^-Gly^9^-Leu^10^-Met^11^-NH_2_	• Cancer growth promotor (astrocytoma, melanoma, neuroblastoma, pancreatic cancer), angiogenesis, migration and metastasis;	[5,49,50]
(Figure 2)	• Study of the synergistic effect of SP and insulin-like growth factor 1 (IGF-1) on corneal epithelial wound healing – synergistic effect possible only in the presence of the SP fragment containing minimum C-terminus 4 amino acids, SP(8–11);	[51]
**[Thi^8^,Met(O_2_)^11^]SP** Pro^4^-Gln^5^-Gln^6^-Phe^7^-Thi^8^-Gly^9^-Leu^10^-Met(O_2_)^11^-NH_2_	Treatment of recurrent and critically located glioblastoma multiforme;	[55]
**[Sar^9^,Met(O_2_)^11^]SP(1–11)** and **(Sendide) [Tyr^6^,D-Phe^7^,D-His^9^]SP(6–11)**	Studies of the role of NK1R in regulation and release of vasopressin peptide;	[56,57]
(X)Arg^1^-Pro^2^-Lys^3^-Pro^4^-Gln^5^-Gln^6^-Phe^7^-Phe^8^-Gly^9^-Leu^10^-Met^11^-NH_2_ ^(1)^ or Arg^1^-Pro^2^-(X)Lys^3^-Pro^4^-Gln^5^-Gln^6^-Phe^7^-Phe^8^-Gly^9^-Leu^10^-Met^11^-NH_2_ ^(1)^	Studies of photoactivatable SP derivatives;	[59]
**Bapa^0^[(pBzl)PheX]SP** ^(2)^ **Bapa^0^[Pro^9^,(pBzl)Hcy(O_2_)^11^]SP** ^(2)^ **Bapa^0^[Hcy(ethylaminodansyl)^11^]SP**	Studies of activation of different second messenger pathways as a result of ligand binding to various NK1Rs sites; studies of dual behaviour of the tested SP derivatives: as antagonists at the NK-1M binding site activating AC pathway or agonists at the NK-1m binding site activating PLC pathway;	[10]
**Septide** [pGlu^6^,Pro^9^]SP(6–11) pGlu^6^-Phe^7^-Phe^8^-Pro^9^-Leu^10^-Met^11^-NH_2_	Agonist as potent as SP in eliciting smooth muscle contraction, however poor competitor of SP due to interaction with another binding site of NK1R (NK-1m, so-called ‘septide-sensitive’);	[7,60]
**GR 73,632** NH_2_(CH_2_)_4_C(O)-Phe^7^-Phe^8^-Pro^9^-(Me)Leu^10^-Met^11^-NH_2_	Approximately 200-fold more potent than SP in inducing the characteristic behavioural response in murine models.	[61]
**Non-mammalian NK1R ligands**		
**Physalaemin** Pyr^1^-Ala^2^-Asp(OH)^3^-Pro^4^-Asp(NH_2_)^5^-Lys^6^-Phe^7^-Tyr^8^-Gly^9^-Leu^10^-Met^11^-NH_2_	Stimulation of extravascular smooth muscles, component of eye drops for Sjögren syndrome treatment and other forms of keratoconjunctivitis sicca;	[3,19,62,63]
**Eledoisin** pGlu^1^-Pro^2^-Ser^3^-Lys^4^-Asp^5^-Ala^6^-Phe^7^-Ile^8^-Gly^9^-Leu^10^-Met^11^-NH_2_	Similar biological activities as Physalaemin but slightly less active and more stable *in vivo*; clinical trials for limb arteriosclerosis treatment; component of eye drops for Sjögren syndrome;	[19,63]
**Sialokinin I** Asn^2^-Thr^3^-Gly^4^-Asp^5^-Lys^6^-Phe^7^-Tyr^8^-Gly^9^-Leu^10^-Met^11^-NH_2_ **Sialokinin II,** Asp^2^-Thr^3^-Gly^4^-Asp^5^-Lys^6^-Phe^7^-Tyr^8^-Gly^9^-Leu^10^-Met^11^-NH_2_	Vasodilation, effect on salivation, influence on the acinar cells of the submandibular glands.	[3,64,65]

^(1)^ X = p-benzoylbenzoic moiety; ^(2)^ Bapa = biotinyl sulfone-5-aminopentanoic acid; Bzl = benzyl; Hcy = homocysteine; HcyO_2_ = homocysteine sulfone.

**Table 2 pharmaceutics-11-00443-t002:** The IC_50_ values determined for SP and various radiobioconjugates based on SP analogues [53].

Substance	IC_50_ ± SEM [nM]
Substance P	2.7 ± 0.22
^111^In-DOTAGA-Substance P	1.1
^111^In-DOTAGA-[Met(O_2_)^11^]-Substance P	9.8 ± 1.00
^111^In-DOTA-[Met(O_2_)^11^]-Substance P	3.55 ± 0.45
^111^In-DOTA-[Sar^9^]-Substance P	3.20 ± 0.30
^111^In-DOTA-[Thi^8^]-Substance P	7.30 ± 2.00
^111^In-DOTA-[Thi^7^]-Substance P	9.40 ± 1.60
^111^In-DOTA-[Sar^9^,Met(O_2_)^11^]-Substance P	2.00 ± 0.00
^111^In-DOTA-[Thi^8^,Met(O_2_)^11^]-Substance P	0.78 ± 0.03
^111^In-DOTA-[Thi^8^,Sar^9^]-Substance P	3.40 ± 0.40
^111^In-DOTA-[Thi^7^,Thi^8^]-Substance P	7.70 ± 0.70

**Table 3 pharmaceutics-11-00443-t003:** Anticancer effects of three most studied NK1R antagonists.

NK1R Antagonists	Anticancer Effect	References
Aprepitant	Tumour cell growth inhibition	[2,16,17,83,84,85,86,87,88,89,90,91,92,93,94,95,96,97,98]
	Tumour cell migration and proliferation inhibition	[2,16,17,83,84,85,86,87,88,89,90,91,92,93,94,95,96,97,98]
L-733,060	Apoptotic action on cells	[2,16,17,85,86,87,88,89,90,91,92,93,94,95,96,97,98]
	Tumour size/volume decrease	[2,86,91,93,94,95,96]
L-732,138	Inflammation state inhibition	[2,93,95,96]
	Angiogenesis decrease	[2,16,86,91,93,95,96]
(Figure 3)	Antiproliferative effect	[2,16,86,91,93,94,95,96]
	Metastases prevention	[2,86,91,93,95,96]

**Table 4 pharmaceutics-11-00443-t004:** Structure and potential application of NK1R diagnostic radioligands based on SP and its analogues or derivatives.

NK1R Radioligand Molecules	Biological Properties and Potential Applications	References
[^125^I]I-BH-[Tyr^8^]SP	• Animal tests: high affinity to pancreatic acinar cells isolated from guinea pigs;	[104]
	• Animal or human tests: used as a radiotracer for determination of specific binding and/or internalisation in various organs, tissues and cells, namely in:	
	▪ anterior pituitary cells	[104,105,106]
	▪ in rat thymus, spleen	[105,106,107]
	▪ brain	[108]
	▪ spinal cord	[109,110]
	▪ chicken small intestine	[111,112,113,114,115]
	▪ epithelial cells	[116]
	▪ mesencephalic primary cultures prepared from embryonic mouse brain	[117,118]
	▪ rat parotid membranes	[119]
	▪ human eyes;	[120]
[^3^H]H-SP	Animal tests: specificity for imaging of cat inflamed bladder tissue;	[121]
[^111^In]In-DTPA-[Arg^1^]SP	• Animal tests: imaging of SP receptor-positive (SPR+) immunologic disorders; high affinity to NK1R presented in parotid gland and brain cortex membranes; rapid enzymatic degradation; high uptake in pancreatic tumour (CA20948), salivary glands, kidneys and arthritic hind leg joints; unable to cross the intact blood–brain barrier;	[122]
	• Clinical trials: used in scintigraphy of immune-mediated diseases;	[123]
[^99m^Tc]Tc-IMB-SP ^(1)^	Animal tests: significant uptake in the salivary glands;	[124]
[^99m^Tc]Tc-Hynic-SP ^(2)^ [^99m^Tc](NS_3_)-Tc-CN-SP and [^99m^Tc]((NS_3_)-Tc-CN)_2_-SP ^(3)^	In vitro study: high stability in biological fluids; relationship between molecular structure and physicochemical properties;	[125]
[^99m^Tc][Tc(N)(Cys-Cys-SP)(PCN)] ^(4)^ [^188^Re][Re(N)(Cys-Cys-SP)(PCN)]	• In vitro study: application of theranostic pair ^99m^Tc and ^188^Re; affinity studies using U87MG cell line expressing NK1R and negative control cell line L-929; • Animal tests: accumulation in salivary glands, kidneys and thymus;	[126]
[^111^In]In-DOTA-[Thi^8^,Met(O_2_)^11^]SP [^68^Ga]Ga-DOTA-[Thi^8^,Met(O_2_)^11^]SP	Clinical trials: used for visualisation of NK1R expression and control of radiocompound distribution at the target site and whole body; administrated simultaneously with therapeutic radiopharmaceutical [^213^Bi]Bi-DOTA-[Thi^8^,Met(O_2_)^11^]SP;	[55,127]
[^99m^Tc][Tc-Hynic-[Tyr^8^,Met(O)^11^]SP ^(5)^	Animal tests: specific uptake in the tumour; stable in HS; internalisation studies on U373 MG astrocytoma cell line; significant accumulation in kidneys;	[128]
[^3^H]H-[Pro^9^]SP and [^3^H]H-propionyl-[Met(O_2_)^11^]SP(7–11)	In vitro study: applied for the studies of different NK1R binding sites: NK-1M (majority) and NK-1m (minority);	[11]
[^125^I]I-BH-[Sar^9^,Met(O_2_)^11^]SP	• Animal tests: comparison of uptake in submandibular gland and in several regions of rat brain of the tested radiocompound and [^125^I]I-BH-SP; • In vitro study: comparison of physicochemical properties of the tested radiocompound and [^125^I]I-BH-SP.	[129,130,131]

**^(1)^** IMB = bifunctional chelator 1-imino-4-mercaptobutyl; **^(2)^** labelling in the presence of EDDA and tricine as coligands; **^(3)^** CN = isocyanide group, NS_3_ = 2,2′,2″-nitrilotriethanethiol; **^(4)^** PCN = tris(2-cyanoethyl)phosphine; **^(5)^** labelling in the presence of EDDA and tricine as coligands.

**Table 5 pharmaceutics-11-00443-t005:** Structure and potential application of NK1R therapeutic radioligands based on SP and its analogues or derivatives.

NK1R Radioligand Molecules	Biological Properties and Potential Applications	Reference
[^177^Lu]Lu-DOTA-SP	• Animal tests: biodistribution studies on mice bearing AR42J pancreatic tumour, high uptake in kidneys, satisfactory uptake in tumour, significant uptake in intestine and stomach; • In vitro study: high specific uptake and internalisation using LN319 cells isolated directly from the tumours;	[133]
[^177^Lu]Lu-DOTAGA-SP [^90^Y]Y-DOTAGA-SP [^213^Bi]Bi-DOTAGA-SP	Medical experiments: well tolerated therapy of critically located gliomas, low toxicity;	[134]
[^90^Y]Y-DOTAGA-SP	Medical experiments: recorded completed encapsulation of the tumour in patients administered with the highest dose;	[135]
[^213^Bi]Bi-DOTA-[Thi^8^,Met(O_2_)^11^]SP [^111^In]In-DOTA-[Thi^8^,Met(O_2_)^11^]SP	Medical experiments: treatment of critically located gliomas; well tolerated and safe for patients; complete necrosis of small tumours and necrosis only in the nearness of the implanted catheters in the case of large tumours;	[55]
[^213^Bi]Bi-DOTA-[Thi^8^,Met(O_2_)^11^]SP [^68^Ga]Ga-DOTA-[Thi^8^,Met(O_2_)^11^]SP	Medical experiments: treatment of patients with secondary GBM (after surgery, chemo- and radiotherapy); very low accumulation in kidneys, urine, bladder and blood; no side effects, necrosis and demarcation of the tumours;	[127]
[^213^Bi]Bi-DOTA-[Thi^8^,Met(O_2_)^11^]SP	Medical experiments: higher efficiency of radiolabelled NK1R ligands application and local brain tumours treatment in patients suffering from secondary GBM compared to standard treatment options;	[136]
[^225^Ac]Ac-DOTA-[Thi^8^,Met(O_2_)^11^]SP	• In vitro study: high affinity to glioblastoma cancer cells: T98G, U87MG, U138MG and glioblastoma stem cells (GSC); significant reduction in glioblastoma cell viability in comparison to the conventional treatment with temozolomide; high cytotoxicity towards GBM stem cells;	[139]
	• Medical experiments: safe and well-tolerated therapy without side effects;	[144]
[^177^Lu]Lu-DOTA-SP(4–11) [^177^Lu]Lu-DOTA-SP(5–11) [^177^Lu]Lu-DOTA-[Thi^8^,Met(O_2_)^11^]SP(5–11)	In vitro study: radiobioconjugates characterized with higher lipophilicity and lower molecular weight than those based on analogue [Thi^8^,Met(O_2_)^11^]SP—changes in physicochemical properties of radiobioconjugates leading to their deeper diffusion into the cavity walls after surgical resection of the tumour.	[145]

**Table 6 pharmaceutics-11-00443-t006:** List of selected radiolabelled NK1R antagonists.

Radiotracer	Structure	Application	Reference
[^11^C]CP-96,345 ^1^	Investigational “lead structure” compounds	Preclinical tests in animal models	[146]
[^11^C]CP-99,994 ^1^			[147]
[^11^C]CP-643,051 ^1^			[148]
[^11^C]GR205171([^11^C]vofopitant) ^2^	Optimized radiotracers	Pharmacodynamics and pharmacokinetics studies, receptor occupancy imaging in clinical trials	[149,151,155,156,157,164,165,166,167,168,169]
[^18^F]SPA-RQ ([^18^F]L-829,165) ^2^			[76,78,150,152,158,159,160,161,162,163]
[^18^F]FE-SPA-RQ			[153,154]
[^11^C]R116301 ^3^			[170,171,172]
[^14^C]aprepitant ^3,4^	Well known high-selective antagonists indicated in CINV	ADME investigations in animals and men	[173]
[^14^C]casopitant ^4^			[174,175]
[^14^C]netupitnat ^4^			[176]

^1^ Structure illustrated in Figure 4; ^2^ Structure illustrated in Figure 5; ^3^ Structure illustrated in Figure 9; ^4^ Structure of nonlabelled compound illustrated in Figure 3.

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
