# Peer review of "The Significance of NK1 Receptor Ligands and Their Application in Targeted Radionuclide Tumour Therapy"

_pharmaceutics, 2019, doi:10.3390/pharmaceutics11090443_

Round 1

Reviewer 1 Report

This excellent review is hampered by many, many grammatical problems.  It will need extensive reworking in order to be read easily.  The frequent issues with syntax and readability reduce the overall impact that the review could have.

Second, the organization of the review needs major revisions.  There are only 3 headings, and headings 2 and 3 are identical ("Significance of NK1 receptor ligands in classical medicine")- Why have two headers that are exactly the same?  Sections 2 and 3 have only two subheaders each and contain large, uninterrupted stretches of text.  It would improve the readability of the article to break these up into additional sections and would give the article better organization.

While the content of the article is outstanding, the issues with grammar and organization are significant.

Author Response

Responses to the comments of Reviewer: 1

Comments and Suggestions for Authors:

This excellent review is hampered by many, many grammatical problems.  It will need extensive reworking in order to be read easily.  The frequent issues with syntax and readability reduce the overall impact that the review could have.

Thank you for this comment. Indeed, there were many inaccuracies in the manuscript (for which we apologize). The whole manuscript has been carefully checked and all mistakes have been removed.

Second, the organization of the review needs major revisions.  There are only 3 headings, and headings 2 and 3 are identical ("Significance of NK1 receptor ligands in classical medicine")- Why have two headers that are exactly the same?  Sections 2 and 3 have only two subheaders each and contain large, uninterrupted stretches of text.  It would improve the readability of the article to break these up into additional sections and would give the article better organization.

Thank you for this comment. We apologize again for the obvious mistake in the heading of part 3. The title should be: "Significance of NK1 receptor ligands in nuclear medicine". Furthermore, according to the Reviewer's suggestion, the whole text of the manuscript has been reworded. 

While the content of the article is outstanding, the issues with grammar and organization are significant.

We used the help of a native speaker and rewrote the manuscript.

Additionally, according to the suggestion of Reviewer 2 all manuscript has been rewritten and in the reviewed version of the manuscript some information are grouped together and presented in the tables.

Reviewer 2 Report

„The significance of NK1 receptor ligands and their application in targeted radionuclide tumour therapy”

1.       List of abbreviations is missing.

2.       The paper is too long and confusing: shortening can be achieved  with  tables and Figures instead of text.

3.       There are several reviews about tachykinins and receptors, therefore the authors should restrict this review to the significance of NKR’s and tachykinin analogues in targeted radionuclide therapy.

Details:

Abstract:

From line 10: The understanding of the function of human substance P (SP) and the overexpression of the neurokinin 1 receptor (NK1R) in various human organs and in cancer cells emphasizes the impact of the SP/NK1R system in health and disease….

Line 25: …subfamily…

Line 29: …C-terminal pentapeptide…

Line 31: …discovered in mammals [REF]…..The compounds listed above…

Fig. 1: use the word „truncated” instead of „short isoform”

Line 40: There are three…..

Line 49-50: „More over, - this sentence should be deleted.

Line 51-64:  The NK1 receptor signal can be transmitted through different intracellular second messenger pathways – this issue has been reviewed by much better articles (see Steinhoff et al., Physiol Rev. 2014; 94).  The story of the second messengers should be illustrated either in a Figure or put in a comprehensive Table. This means that lines 51-64 should be deleted.

Lines 65-69: The aim of the present review is to discuss the literature data about the chemical and biological aspects of natural and synthetic NK1R ligands used in classical and nuclear medicine, especially targeted radionuclide therapy.

Line 70: what is classical medicine in this context? It would be better to delet this and use simply the subheading: AGONIST LIGANDS OF NK1 RECEPTOR

Line 72: The endogenous peptide ligands of the NK1R are the tachykinins, a large family of neuropeptides, produced by neuronal and glial cells.

Lines 88-90: delete these sentences.

Figure 3 does not have too much impact: there are no explanations of colors, or indication of the extra- and intracellular surfaces.

Line 97-98:  …plasma SP is stable for hours. In the biodegradation process….

Line 102: …responsible for the neural-immune/hematopoietic cross-talk [REF]

Line 106:-----„The induced…signals” sentence should be deleted. The next sentence should be deleted, too: „The magnitude of these…stimuli. Mantyh’s studies of pharmacologic or genetic inactivation of NK1 receptors showed that : delete this. Continue like this:  The inhibition of NK pathway (what is NK pathway?) may be a useful treatment of depression and associated anxiety. The blocking of NK1R can prevent vomiting after anesthesia or chemotherapy.

Lines 116-129: the data cited in these lines should be better in a table.

The entire subheading (lines 72-209) is about the agonists: better list the agonists (natural and synthetic), describe their exact chemical structure, and then put them into a table, which describes their biological actions and medical usefulness. The present description is boring and confusing. With a table you will be more specific, shorter, concise and understandable.

The same critical comment is valid for the ANTAGONISTS. The chemical formulas of the antagonists are well presented.

Line 213: the first sentence should be deleted. Begin with: „Despite the wide expression of the NK1R and the implication of SP in physiological regulation, the NK1R antagonists are still rarely applied in clinical routine. (delete: „non-derivatives of SP” what is this??) First of them, aprepitant….

The rest of text is OK – down to line 249.

Lines 258-260: delete the sentence: „Despite the fact that in each…..clinical efficacy.”

Line 261: On the other hand, great expectations are arising from the application of NK1R antagonists for the targeted treatment of malignant tumours. The NK1R ligands were investigated in terms…. I think that in this paragraph a table would be very useful, much better than the descriptive text.

Line 285: this subtitle is the repetition of the previous (line 70)

The real title should be: SP analogues for targeted radionuclide tumour therapy

In this paragraph, first  we need a concise description of the chemical structure and the different isotope-labelling methods. Then we need the description of the binding by different tissues (in vivo and in vitro), the methods through which the chemicals enter the organs, visualization of the binding, degradation and excretion of the compounds. Again, a table could be helpful, because this text is very difficult to follow. We need some text, but separate tables may contain results in animals and results in human patients obtained with different radionuclides. As the substances are carefully listed in Table 1, besides the IC50 values, the authors should list also the other parameters of the experiments (tissue, tumour cell type, etc…)

Lines 290-303: „In order to allow the labelling of SP with iodine-125, the amino acid Phe in the position 8 was replaced by Tyr. The [Tyr8]SP was radioiodinated  with  125I by the Bolton-Hunter agent………The results indicated high affinity of SP to all listed cells and organs.

Line 316: why IC50 values? we are talking about agonists

Line 318:…pointing to renal excretion…elimination.

Line 319:…At 30 min after the injection…

Line 321: …Rapid enzymatic degradation of the tested radiocompound was detected [REF] which resulted in an approx. 3 min biological half-life in the blood.

Line 334: Avoid using „The group from Erasmus University Rotterdam…”, or mentioning the author names (e.g.: Breeman et al.), or mentioning the date of the experiment:  just put the reference number, and the reader will know it. These details are unnecessary, and make the text longer.

Lines 288-585: in this section the effectiveness of isotope-labelleled SP analogues (NK1R agonists) is discussed, which is one of the main topics of this review. However, the text is difficult to follow. We recommend didactic subdivisions of the text with subtitles; e.g.: subdivisions according to the kind of isotope labelling, subdivisions according to the type of the tumour, subdivisions according the type of experiment (tissue culture, animal, human). With these subdivisions, the advantages and disadvantages of the compounds could have been discussed, as well – giving a useful information for the reader. Tables are also useful – something like Table 2 listing the antagonists, the different isotope labels, the application types and references.

Line 594: „An idea of application of neurokinin 1 receptor antagonists for external imaging of the brain arise with reports mainly about their presumed antidepressant application.” This sentence is without sense (there are several others without sense in the text).

Line 597: ….radiotracers made at the end of XX – what is this?

In the case of the antagonists we need a clear list, naming of the structures to which they bind and give signal (uptake signal), clear applications, indications (diseases, therapy or diagnosis), absorption, distribution, metabolism and excretion (if data are available).

Summarizing the above critical notes:

1.       Thorough  english language edition is necessary.

2.       The shortening of the text (by approx. 30%) is necessary.

3.       New headings (e.g.: 1. Introduction; 2. Structure of NK1 receptor agonists; 3. Structure of NK1 receptor antagonists; 4. Structure of SP analogues used in targeted radionuclide tumour therapy; 5. Structure of NK1 receptor antagonists used for targeted radionuclide therapy; 6. Conclusions – at present it is missing) are necessary.

4.       List of abbreviations is needed.

Author Response

Responses to the comments of Reviewer: 2

„The significance of NK1 receptor ligands and their application in targeted radionuclide tumour

therapy”

List of abbreviations is missing.

Thank you for this remark. The abbreviation list has been completed in the manuscript.

The paper is too long and confusing: shortening can be achieved with tables and Figures

instead of text.

Thank you for the remark. The manuscript text has been shortened.

There are several reviews about tachykinins and receptors, therefore the authors should restrict this review to the significance of NKR’s and tachykinin analogues in targeted radionuclide therapy.

Thank you for this comment. We considered this remark in improved manuscript version.

Details:

Abstract:

From line 10: The understanding of the function of human substance P (SP) and the overexpression

of the neurokinin 1 receptor (NK1R) in various human organs and in cancer cells emphasizes the

impact of the SP/NK1R system in health and disease….

The sentence has been replaced with the proposed version.

Line 25: …subfamily…

The error has been removed.

Line 29: …C-terminal pentapeptide…

The error has been removed.

Line 31: …discovered in mammals [REF]…..The compounds listed above…

We have added the adequate reference and corrected the sentence.

Fig. 1: use the word „truncated” instead of „short isoform”

Thank you for the remark – the term ‘short isoform’ has been replaced by ‘truncated isoform’.

Line 40: There are three…..

The sentence has been corrected.

Line 49-50: „More over, - this sentence should be deleted.

The sentence has been deleted.

Line 51-64: The NK1 receptor signal can be transmitted through different intracellular second

messenger pathways – this issue has been reviewed by much better articles (see Steinhoff et al.,

Physiol Rev. 2014; 94). The story of the second messengers should be illustrated either in a Figure or

put in a comprehensive Table. This means that lines 51-64 should be deleted.

Thank you for this remark – this brief and really insufficient information about the NK1 receptor signal transmitted through different intracellular second messenger pathways has been deleted from the manuscript.

Lines 65-69: The aim of the present review is to discuss the literature data about the chemical and

biological aspects of natural and synthetic NK1R ligands used in classical and nuclear medicine,

especially targeted radionuclide therapy.

According to the Reviewer’s remark the sentence has been changed in the manuscript.

Line 70: what is classical medicine in this context? It would be better to deleted this and use simply the subheading: AGONIST LIGANDS OF NK1 RECEPTOR

Thank you for the comment – we have changed the subheading and generally rewritten the manuscript.

Line 72: The endogenous peptide ligands of the NK1R are the tachykinins, a large family of

neuropeptides, produced by neuronal and glial cells.

Thank you for the comment - the text has been changed.

Lines 88-90: delete these sentences.

We apologize for this error. This sentence remained by our inattention and has already been removed.

Figure 3 does not have too much impact: there are no explanations of colors, or indication of the

extra- and intracellular surfaces.

Thank you for the comment – this figure did not provide any information indeed and was removed.

Line 97-98: …plasma SP is stable for hours. In the biodegradation process….

Thank you for the comment - the text has been corrected.

Line 102: …responsible for the neural-immune/hematopoietic cross-talk [REF]

The missing reference has been completed.  

Line 106:-----„The induced…signals” sentence should be deleted. The next sentence should be

deleted, too: „The magnitude of these…stimuli. Mantyh’s studies of pharmacologic or genetic

inactivation of NK1 receptors showed that : delete this. Continue like this: The inhibition of NK

pathway (what is NK pathway?) may be a useful treatment of depression and associated anxiety. The

blocking of NK1R can prevent vomiting after anesthesia or chemotherapy.

Thank you for the comment - the text has been changed.

Lines 116-129: the data cited in these lines should be better in a table.

Thank you for the comment –  selected data are included in the table.

The entire subheading (lines 72-209) is about the agonists: better list the agonists (natural and

synthetic), describe their exact chemical structure, and then put them into a table, which describes

their biological actions and medical usefulness. The present description is boring and confusing. With

a table you will be more specific, shorter, concise and understandable.

Thank you for your valuable comments. The whole manuscript has been rewritten and most of the text has been transferred into the table.

The same critical comment is valid for the ANTAGONISTS. The chemical formulas of the antagonists

are well presented.

We corrected the text of the manuscript according to the Reviewer's suggestion.

Line 213: the first sentence should be deleted. Begin with: „Despite the wide expression of the NK1R

and the implication of SP in physiological regulation, the NK1R antagonists are still rarely applied in

clinical routine. (delete: „non-derivatives of SP” what is this??) First of them, aprepitant….

The rest of text is OK – down to line 249.

We have corrected the text according to the Reviewer’s remark.

Lines 258-260: delete the sentence: „Despite the fact that in each…..clinical efficacy.”

The sentence has been deleted.

Line 261: On the other hand, great expectations are arising from the application of NK1R antagonists

for the targeted treatment of malignant tumours. The NK1R ligands were investigated in terms…. I

think that in this paragraph a table would be very useful, much better than the descriptive text.

According to the Reviewer’s remark the paragraph has been changed .

Line 285: this subtitle is the repetition of the previous (line 70)

The real title should be: SP analogues for targeted radionuclide tumour therapy

This mistake arose as a result of our inattention again. The planned title was: Significance of NK1 receptor ligands in nuclear medicine. In the reviewed manuscript we suggest slightly changed work arrangement and slightly changed subtitles.

In this paragraph, first we need a concise description of the chemical structure and the different

isotope-labelling methods. Then we need the description of the binding by different tissues (in vivo

and in vitro), the methods through which the chemicals enter the organs, visualization of the binding,

degradation and excretion of the compounds. Again, a table could be helpful, because this text is

very difficult to follow. We need some text, but separate tables may contain results in animals and

results in human patients obtained with different radionuclides. As the substances are carefully listed

in Table 1, besides the IC50 values, the authors should list also the other parameters of the

experiments (tissue, tumour cell type, etc…)

Thank you for this critical, but also constructive remark. We have rewritten the manuscript, the issues discussed are systematically grouped and some data are included in the tables. According to the data in the Table 1 (in the reviewed manuscript in the Table 2) – these data include only IC50 values because the aim of this study (Merlo, A.; Mäcke, H.; Reubi, J-C.; Good, S. Radiolabeled conjugates based on substance P and the uses thereof. Patent: International application number: PCT/EP2004/050329; International publication number: WO 2004/082722.) was to select the SP analogue/derivative characterized with the highest stability and affinity to NK1R only.

According to the Reviewer’s suggestion we added in the rewritten text of manuscript the tables containing concise information about the content of the cited works.

Lines 290-303: „In order to allow the labelling of SP with iodine-125, the amino acid Phe in the

position 8 was replaced by Tyr. The [Tyr8]SP was radioiodinated with 125I by the Bolton-Hunter

agent………The results indicated high affinity of SP to all listed cells and organs.

Thank you for the remark – the relevant fragment of manuscript has been changed according to the Reviewer's comments.

Line 316: why IC50 values? we are talking about agonists

In the manuscript was: “The in vitro binding and autoradiographic experiments performed on parotid gland and brain cortex membranes as well as on 10-μm slices of the submandibular gland of rat, demonstrated high affinity of [111In]In-[DTPA-Arg1]SP to NK1 receptors. The IC50 values were in the nanomolar range.” (In the reviewed manuscript this information is placed in Table 2). Indeed, SP is a very strong agonist to NK1 receptor. However, the aim of the study reported in the cited work was to test receptor affinity of novel ligands (among others the conjugate DTPA-[Arg1]SP). This study was performed in competition binding assay using 125I-(Bolton Hunter)-SP as radioligand and  [DTPA-Arg1]SP as competitor. In such experiments one of the compounds always acts as an inhibitor against the other one and the experimentally determined parameter is IC50 (the concentration of the inhibitor required to inhibit the binding of a radioligand by 50%) [Lutea A.A. de Jong, Donald R.A. Uges, Jan Piet Franke, Rainer Bischoff, “Receptor–ligand binding assays: Technologies and Applications”, Journal of Chromatography B, 829 (2005) 1–25.;  Edward C. Hulme, Mike A. Trevethick, “Ligand binding assays at equilibrium: validation and interpretation”, British Journal of Pharmacology (2010), 161, 1219–1237.]

Line 318:…pointing to renal excretion…elimination.

The sentence has been changed.

Line 319:…At 30 min after the injection…

Thank you for the remark – the sentence has been corrected.

Line 321: …Rapid enzymatic degradation of the tested radiocompound was detected [REF] which

resulted in an approx. 3 min biological half-life in the blood.

The sentence has been changed.

Line 334: Avoid using „The group from Erasmus University Rotterdam…”, or mentioning the author

names (e.g.: Breeman et al.), or mentioning the date of the experiment: just put the reference

number, and the reader will know it. These details are unnecessary, and make the text longer.

Thank you for that remark. Some of these phrases have been corrected, but in our opinion some contain useful details and help the recipient to follow specific reference or distinguish that specific one from other.

Lines 288-585: in this section the effectiveness of isotope-labelleled SP analogues (NK1R agonists) is

discussed, which is one of the main topics of this review. However, the text is difficult to follow. We

recommend didactic subdivisions of the text with subtitles; e.g.: subdivisions according to the kind of

isotope labelling, subdivisions according to the type of the tumour, subdivisions according the type of

experiment (tissue culture, animal, human). With these subdivisions, the advantages and

disadvantages of the compounds could have been discussed, as well – giving a useful information for

the reader. Tables are also useful – something like Table 2 listing the antagonists, the different

isotope labels, the application types and references.

According to the Reviewer’s suggestion we added in the rewritten text of manuscript the tables containing concise information about the research discussed in our work.

Line 594: „An idea of application of neurokinin 1 receptor antagonists for external imaging of the

brain arise with reports mainly about their presumed antidepressant application.” This sentence is

without sense (there are several others without sense in the text).

The sentence has been deleted.

Line 597: ….radiotracers made at the end of XX – what is this?

The sentence has been corrected.

In the case of the antagonists we need a clear list, naming of the structures to which they bind and

give signal (uptake signal), clear applications, indications (diseases, therapy or diagnosis), absorption,

distribution, metabolism and excretion (if data are available).

Thank you for that suggestion.

Summarizing the above critical notes:

Thorough english language edition is necessary. The shortening of the text (by approx. 30%) is necessary. New headings (e.g.: 1. Introduction; 2. Structure of NK1 receptor agonists; 3. Structure of

NK1 receptor antagonists; 4. Structure of SP analogues used in targeted radionuclide tumour

therapy; 5. Structure of NK1 receptor antagonists used for targeted radionuclide therapy; 6.

Conclusions – at present it is missing) are necessary.

List of abbreviations is needed.

Thank you for your effort in whole careful revision. Whole text has been precisely corrected and shortened, written language was proofread and edited. Thank you for division remark with new headings proposal. We have imply that change with our heading statements. However, we considered that separate conclusion section is redundant and it will extend excessively an article. Instead, we propose brief sum up (2-4 sentences) at the end of each paragraph. Obligatory abbreviation list has been added.

Reviewer 3 Report

The review is an interesting one and is based on the consultation of works from 1980 to 2019. Is of interest to researchers in the field of synthesis of novel compounds that have the therapeutic target of NK1 receptor. The work deserves to be published but requires the renumbering of bibliographic titles and some corrections in the text:

careful references 10,11

597: of the 20th century

Figure 10 structure aprepitant: the C is tetravalent

Author Response

Responses to the comments of Reviewer: 3

Comments and Suggestions for Authors:

The review is an interesting one and is based on the consultation of works from 1980 to 2019. Is of interest to researchers in the field of synthesis of novel compounds that have the therapeutic target of NK1 receptor. The work deserves to be published but requires the renumbering of bibliographic titles and some corrections in the text:

Thank you for the comments. The entire list of cited works as well as the whole text of manuscript have been checked and corrected.

careful references 10,11

The references 10 and 11 have been corrected (the reference 11 has been completed).

597: of the 20th century

The sentence has been corrected.

Figure 10 structure aprepitant: the C is tetravalent

That is true, it is tetravalent. Structures from Figure 10 are drawn in sticks format without hydrogen atoms and with highlighted C-atoms of carbon-14. That is why aprepitant carbon-14 seems like trivalent because is drawn in sticks format. In our opinion that structure format is more transparent and often applied in labelled compound structures (e.g. J Label Compd Radiopharm 2004; 47: 837–846.)

Additionally, according to the suggestion of Reviewer 2 all manuscript has been rewritten and in the reviewed version of the manuscript some information are grouped together and presented in the tables.

Round 2

Reviewer 1 Report

There are still a few places were grammar and spelling need to be corrected.  However the manuscript is greatly improved, and the few minor changes can be easily made.

Author Response

Please find enclosed all responses in the pdf manuscript, while all changes are marked in gray in word. 

Reviewer 2 Report

The corrections made by the authors are satisfactory and acceptable.

Author Response

Dear Reviewer,

Thank you very much for accepting all changes in our manuscript.

A.Majkowska-Pilip